# KAPALM: Knowledge grAPh enhAnced Language Model for Fake News Detection

**Jing Ma[1,2]   Chen Chen[1,2*]   Chunyan Hou[3]   Xiaojie Yuan[1,2]**

[1] College of Computer Science, Nankai University, Tianjin, China
[2] MoE Key Lab of DISSec, Nankai University, Tianjin, China
[3] School of CSE, Tianjin University of Technology, Tianjin, China
majing@mail.nankai.edu.cn, {nkchenchen,yuanxj}@nankai.edu.cn,
houchunyan@tjut.edu.cn

## Abstract

Social media has not only facilitated news consumption, but also led to the wide spread of fake news. Because news articles in social media are usually condensed and full of knowledge entities, existing methods of fake news detection use external entity knowledge to improve the effectiveness. However, the majority of these methods focus on news entity information and ignore the structured relation knowledge among news entities. To address this issue, in this work, we propose a Knowledge grAPh enhAnced Language Model (KAPALM) which is a novel model that fuses coarse- and fine-grained representations of entity knowledge from Knowledge Graphs (KGs). Firstly, we identify entities in news content and link them to entities in KGs. Then, a subgraph of KGs is extracted to provide structured relation knowledge of entities in KGs and fed into a graph neural network to obtain the coarse-grained knowledge representation. This subgraph is pruned to provide fine-grained knowledge and fed into the attentive graph pooling layer. Finally, we integrate the coarse- and fine-grained entity knowledge representations with the representation of news content for fake news detection. The experimental results on two benchmark datasets show that our method is superior to state-of-the-art baselines in the full-scale setting. In addition, our model is competitive in the few-shot setting.

## 1 Introduction

In recent years, with the development of the Internet, social media such as Facebook and Twitter have become the main platforms for people to consume news due to their real-time and easy access. However, social media is a double-edged sword and it enables people to be exposed to fake news. The wide spread of fake news can misguide public opinion, threaten people's health, and even cause devastating effects on society (Vosoughi et al., 2018).

---

[*] Corresponding author.

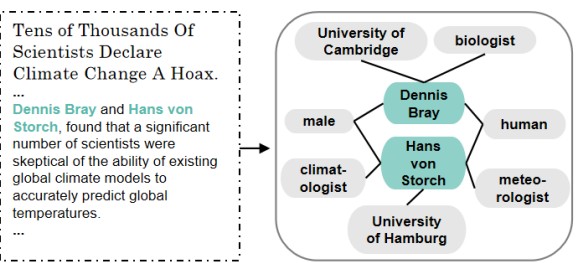

Figure 1: A news article on PolitiFact and its entity graph.

Thus, the research on automatic fake news detection is desirable.

Prior researches are based on traditional machine learning methods and hand-crafted features based on news content to combat fake news on social media (Castillo et al., 2011;Kwon et al., 2013;Zubiaga et al., 2017;Przybyla, 2020;). To avoid manual features, deep neural networks models such as Convolutional Neural Networks (CNN) and Recurrent Neural Network (RNN), have been used to learn the high-level feature representations automatically and achieve great performance in detecting fake news (Ma et al., 2016; Wang, 2017; Shu et al., 2019). For the past few years, Pre-trained language models (PLMs), such as BERT (Devlin et al., 2018) and RoBERTa (Liu et al., 2019), have been the mainstream approaches which provide the contextualized representation of textual content in natural language processing field and also achieve the competitive performance for fake news detection (Pelrine et al., 2021; Sheng et al., 2021). However, these works fail to consider the knowledge entities in news articles. Although news content is condensed and full of knowledge entities by which people usually verify the veracity of news content, PLMs are not effective in capturing the knowledge-level relationship between entities. As shown in the left of Figure 1, it is hard to detect the veracity of news article exactly without the entity knowledge about *Dennis Bray* and *Hans von Storch*.

Large Knowledge Graphs (KGs), such as Freebase (Bollacker et al., 2008) and Wikidata (Vrandecic and Krötzsch, 2014), contain a large number of structured triplets, which can serve as the entity knowledge base for detecting fake news. Existing researches have demonstrated the significant role of KGs (Pan et al., 2018; Dun et al., 2021; Jiang et al., 2022). Specifically, Dun et al. (2021) proposed knowledge attention networks to measure the importance of knowledge and incorporate both semantic-level and knowledge-level representations of news content. Jiang et al. (2022) proposed a knowledge prompt learning method which incorporated KGs into the prompt representation to make it more expressive for verbal word prediction. Hu et al. (Hu et al., 2021) explored the structural entity embedding and compared contextual entity representations with the corresponding entity representations in the knowledge graph. However, they focus on the textual representation of entities in KGs and have not explored the in-depth structured knowledge associated with the new article. As shown in Figure 1, we construct a subgraph of KGs for the news article which is named as *entity graph* throughout this paper, and this entity graph is helpful to detect fake news with the knowledge that "Dennis Bray is a biologist and Hans von Storch is a climatologist and meteorologist".

In this paper, we proposed a Knowledge grAPh enhAnced Language Model (KAPALM) which is enabled to explore both coarse- and fine-grained knowledge in KGs for fake news detection. Firstly, we identify entities from news content and link them to the large-scale KGs. An entity graph is constructed by extracting the first-order neighbors of identified entities, and fed into a Graph Neural Network (GNN) to obtain coarse-grained structured knowledge representations. Then, we prune the entity graph and feed it into the attentive graph pooling layer for fine-grained knowledge representations. Finally, we integrate coarse- and fine-grained knowledge representations with the representation of news content encoded by PLMs for fake news detection. The primary contributions of this paper can be concluded as follows:

- We propose a method to construct an entity graph for each news article. Our method is enabled to extract coarse-grained structured knowledge from the entity graph and provide fine-grained knowledge by the pruned entity graph.

- We propose a Knowledge grAPh enhAnced Language Model (KAPALM) to integrate the coarse- and fine-grained knowledge representations with the representation of news content for fake news detection.

- We compare our model with the state-of-the-art methods on two benchmark datasets in few-shot and full-scale settings. The experimental results demonstrate that our model outperforms the state-of-the-art methods in the full-scale setting. Moreover, our model is competitive in the few-shot setting.

## 2 Related Work

### 2.1 Fake News Detection

In this section, we will briefly introduce the related work on fake news detection. This work (Zhou and Zafarani, 2021) outlines early methods which design handcrafted features and utilize statistical machine learning methods to detect the authenticity of a given news. These work (Castillo et al., 2011; Przybyla, 2020; Kwon et al., 2013; Zubiaga et al., 2017) proposed to utilize statistical information to leverage textual and social features to detect fake news. Also, some works (Ajao et al., 2019; Zhang et al., 2021) focus on capturing sentiment features to better detect fake news. Next, we will explore relevant research on the application of deep learning techniques in the task of detecting fake news. With the development of deep learning techniques, some models such as CNN, RNN, and Transformer(Kim, 2014; Liu et al., 2016; Vaswani et al., 2017) are being used for fake news detection(Dun et al., 2021; Ma et al., 2016; Samadi et al., 2021; Shu et al., 2019; Jiang et al., 2022). Dun et al.(2021) proposed a approach which applied a knowledge attention network to incorporate external knowledge based on Transformer. Jiang et al.(2022) proposed KPL which incorporated prompt learning on the PLMs for the first time and enriched prompt template representations with entity knowledge and achieved state-of-the-art performance on two benchmark datasets, PolitiFact(Shu et al., 2020) and Gossipcop(Shu et al., 2017).

### 2.2 Knowledge Graphs

External knowledge can provide necessary supplementary information for detecting fake news. It is stored in a knowledge graph format, which contains information about entities in the real world,

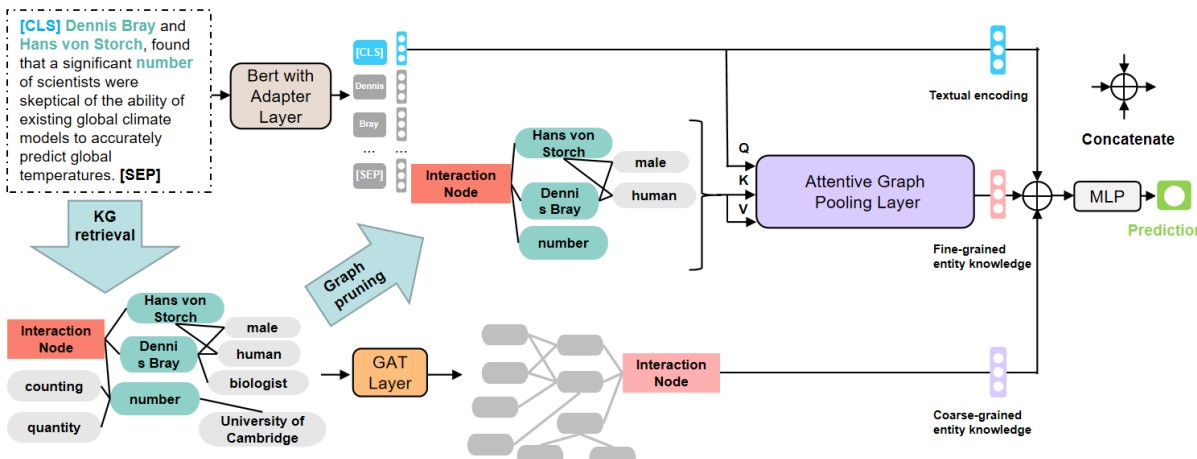

Figure 2: The overview of our proposed model KAPALM

such as people, places, etc. Nowadays, knowledge graphs are used in various natural language processing applications, such as news recommendation(Wang et al., 2018), fact verification(Zhong et al., 2020;Zhou et al., 2019), and fake news detection(Dun et al., 2021;Jiang et al., 2022). Among them, Dun et al.(2021) designed two attention mechanisms of news towards entities and news towards entities and entity context to capture the different importance of entities in detecting fake news. Jiang et al.(2022) utilized a prompt template and integrated entities extracted from news articles into the prompt representation for fake news detection. Hu et al. (Hu et al., 2021) proposed a novel graph neural model, which compared the news to the knowledge graph through entities for fake news detection. The limitations of the these methods are that they all focus on news entity information and ignore the structured information of KGs associated with the news article. Our proposed model can learn knowledge from large-scale PLMs, acquire entity and topological structure information from external KGs, and improve the performance of fake news detection.

## 3 Model

We illustrate the framework of KAPALM in Figure 2. The input of KAPALM consists of the news content and the subgraph constructed from KGs. The output of the model is the label for binary classification of fake news detection. First, for each piece of news, we use a pre-trained language model with the adapter layer (Pfeiffer et al., 2020) to encode the text of news content. Next, entity linking is used to identify entities in new content and we extract

these entities and their corresponding first-order neighbors in the knowledge graph to construct the coarse-grained entity graph of the news. Entity extraction and entity graph construction are described in the knowledge extraction section. Then, we feed the constructed entity graph into a graph attention network (Velickovic et al., 2017) to generate a new entity graph that incorporates both the knowledge graph and entity information. We use the interaction node in the entity graph to represent coarse-grained entity knowledge. Furthermore, to obtain fine-grained entity knowledge representation, we prune the constructed entity graph and then use an attention mechanism for graph pooling. Finally, we concatenate the representations of the news content, coarse- and fine-grained entity knowledge representation, and feed them into a fully connected neural network to obtain the prediction for fake news detection.

### 3.1 Text Encoder

This module is to generate the representation of the news content. We utilize a pre-trained language model with Adapter (Pfeiffer et al., 2020) to encode the news content for capturing the semantic information contained in the news article. The pre-trained language model is usually pre-trained on a large amount of textual corpus, and is able to capture the semantics of the news content. However, when the pre-training task is not similar to the downstream task, fine-tuning pre-trained language model is required to achieve the state-of-the-art performance on the downstream task. In addition, the large number of parameters in pre-trained language model can lead to the high time and space

costs during the full fine-tuning. Thus, adapter turning is used as a lightweight fine-tuning strategy that achieves competitive performance to full fine-tuning on downstream tasks. Adapters add a small set of additional parameters at every layer of Transformer (Vaswani et al., 2017). The parameters of pre-trained language model are kept frozen, and the parameters of adapters are trained during fine-tuning. The adapter layer can be represented as the following equation:

$$h = h + W_{up}ReLU(hW_{down}) \qquad (1)$$

where $W_{up}$ and $W_{down}$ are parameter matrixes.

### 3.2 Knowledge Encoder

#### 3.2.1 Entity Graph Construction and Pruning

This module is used to extract relevant entities from the knowledge graph and construct two subgraphs. Figure 3 illustrates the pipeline process which includes entity linking, entity graph construction, and entity graph pruning. Firstly, we use entity linking (Milne and Witten, 2008; Sil and Yates, 2013) to extract the entities mentions from the news content, align them with entities in the knowledge graph, and obtain their first-order neighbors in the knowledge graph. As a result, we obtain the entity set $E = \{e_1, e_2, ..., e_n\}$ and its first-order neighbor set $N = \bigcup_{e_i \in E} N_{e_i}$, where $N_{e_i}$ is the first-order neighbor set of $e_i$. Secondly, we construct the entity graph $G$. In order to better aggregate entity graph information, we create an interaction node that is connected to every $e_i \in E$, and then connect each $e_i$ to its corresponding first-order neighbors in $N_{e_i}$.

Due to the large number of entities in the constructed entity graph $G$, the information associated with relevant entities in news content cannot be effectively captured. Therefore, we prune the entity graph to retain only the neighbors on one path of a pair of entities. In other words, we remove the first-order neighbors with a degree of one in the entity graph. In this case, we obtain the pruned entity graph $G'$, which consists of all entities in $E$ and their remained first-order neighbors $N' = \bigcup_{e_i \in E} N'_{e_i}$, where $N'_{e_i}$ is remained first-order neighbors of $e_i$.

#### 3.2.2 Coarse-grained Knowledge

In this module, we aim to obtain coarse-grained knowledge representations for entities in the entity graph. After the construction of the entity graph, we use PLMs to initialize the representations of nodes. While the representation of the interaction node is initialized with the embedding of [CLS] token for the news content, the initialization of other entity nodes is provided by encoding the text of the entity name. As shown in the bottom of Figure 2, the initialized entity graph is fed into a Graph Attention Network (Velickovic et al., 2017) to aggregate the information among all entities. The representation of interaction node is enabled to integrate entity knowledge with the contextual representation of news content. Because the entity graph includes many noisy entities, we input only the representation of interaction node which is named the coarse-grained knowledge representation. The hidden representation of coarse-grained knowledge $a$ is calculated as follows:

$$a = Interaction(GAT(G)) \qquad (2)$$

where $G$ denotes the entity graph and function $Interaction$ returns the representation of the interaction node.

#### 3.2.3 Fine-grained Knowledge

After pruning the entity graph, we feed the pruned graph into the attentive graph pooling layer to extract the fine-grained entity knowledge representation. Entities in the entity graph, especially those first-order neighbor entities, do not have the same role in detecting the veracity of news articles. Therefore, we propose to utilize the attentive graph pooling layer which is based on the multi-head attention mechanism to measure the importance of different entities to fake news detection. The output of the attentive graph pooling layer is called the fine-grained knowledge representation.

**Attentive Graph Pooling Layer** As shown in Figure 2, the query is the hidden state of the news article encoded by PLMs, while both the key and value have the same representation which is derived from the hidden states of $e_i \in E \bigcup N'$ where $E$ and $N'$ denote the entity set and the remained neighbour set respectively. Each entity is assigned a corresponding weight by calculating the similarity between the news and this entity. The attention formula is shown as follows:

$$Q = W_Q h, K = W_K c, V = W_V c \qquad (3)$$

$$s = Attention(Q, K, V) = Softmax(\frac{QK^T}{\sqrt{d_k}})V \qquad (4)$$

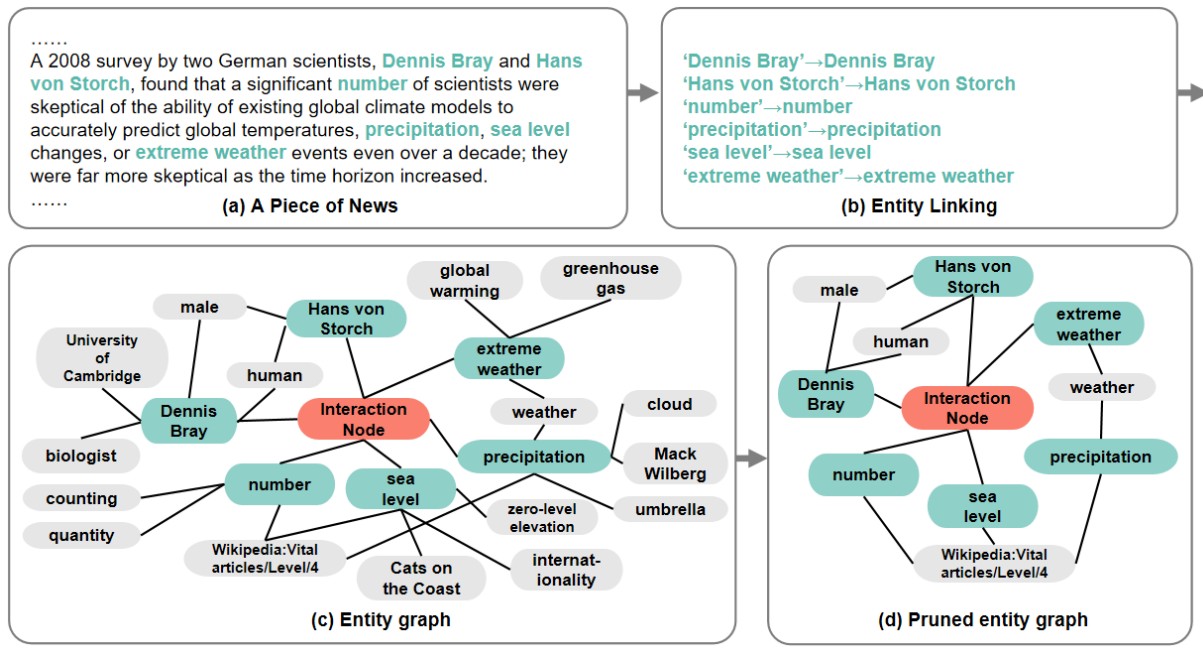

Figure 3: The process of knowledge extraction.

where $h$ and $c$ denote the hidden states of a news article and an entity in $E \bigcup N'$ respectively, and $W_Q$, $W_K$ and $W_V$ are parameter matrices.

### 3.3 Knowledge Fusion

#### 3.3.1 Deep Neural Network Classifier

We concatenate the representations of the news content $h$, coarse-grained knowledge representation $a$, and fine-grained knowledge representation $s$ to obtain the final representation $z$. Then, $z$ is fed into a fully connected layer followed by a softmax function to predict the probability distribution $P$ of the news article's label.

$$z = Concat(h, a, s) \qquad (5)$$

$$P = Softmax(W_o z + b_o) \qquad (6)$$

The model is trained to minimize the following cross-entropy loss function.

$$J = -\sum_{i \in T} Y_i log(P_i) + \frac{\lambda}{2}||\Theta||^2 \qquad (7)$$

where $T$ refers to the training dataset, $P_i$ and $Y_i$ denote the distributions of the prediction and true label of the sample $i$, $\lambda$ denotes the coefficient of $L2$ regularization, and $\Theta$ denotes the model parameters.

### 4 Experiments

#### 4.1 Datasets

To evaluate the proposed model, we conduct experiments on two datasets *PolitiFact* and *Gossipcop* that are included in a benchmark datasets called FakeNewsNet (Shu et al., 2020). The detail of these news datasets are shown in Table 1. We study our proposed model in both few-shot and full-scale settings.

**Few-shot settings** For the purpose of replicating low-resource situations in real-world scenarios, we randomly select $k \in (2, 4, 8, 16, 100)$ news articles as the training set and create the validation set of the same size. The other news articles are used as the test set. We follow (Jiang et al., 2022), and sample the few-shot data by 10 random seeds and use the average value calculated after deleting the maximum and minimum scores as the final score.

**Full-scale settings** For a dataset, we reserve 10% of the news articles as the validation set, and 5-fold cross validation is conducted on the other new articles. Finally, the average score is reported.

#### 4.2 Implementation Details

TagMe (Ferragina and Scaiella, 2010) is adopted to extract knowledge mentions from news articles. We utilize the BERT-base version with adapter as the pre-trained language model (Pfeiffer et al., 2020) to extract text features, which is based on the HuggingFace Transformer Library (Wolf et al.,

| Statistics | Politifact | Gossipcop |
|---|---|---|
| # True news | 442 | 9714 |
| # Fake news | 371 | 4415 |
| # Total news | 813 | 14129 |
| avg.# words/news | 1449 | 576 |
| avg.# entities/news | 55 | 29 |

Table 1: Statistics of the new datasets. "#" and "avg.#" denote "the number of" and "the average number of".

2020). For the coarse-grained entity knowledge, we use GAT (Velickovic et al., 2017) as our graph neural network model. In the attentive graph pooling layer, we set the attention head to 2. The size of hidden layer is set to 200 and the dropout rate to 0.2 in the MLP layer. Adam is used to optimize the model's parameters in the training, The learning rate is 1e-5 and the batch size is 10 for training model.

Because our aim focuses on detecting fake news, fake news articles are regarded as positive examples and F1-score (F1) is used as the evaluation metric to measure the classification performance.

### 4.3 Baselines

We compare our proposed model with the following baselines:

(1) **DTC** (Castillo et al., 2011): DTC is the decision tree model, which detects the authenticity of news by utilizing hand-crafted features.

(2) **RFC** (Kwon et al., 2013): RFC is the random forest classifier based on hand-crafted features to detect whether news is true or fake.

(3) **SVM** (Yang et al., 2012): SVM denotes a classification model that uses a hyperplane to separate news into true and fake news in high dimensional feature space.

(4) **TextCNN** (Kim, 2014): TextCNN is a popular deep learning model for text classification, which applies convolutional filters with various window sizes to extract text features. These features are fed into pooling layer to furtherly capture the most salient features to judge whether news is true or fake.

(5) **BiLSTM** (Bahad et al., 2019;Hochreiter and Schmidhuber, 1997): BiLSTM denotes the bidirectional long short-term memory which introduces a two-directional recurrent netural

network architecture to better capture the temporal dependencies in text data. We utilize it to judege news's authenticity.

(6) **KCNN** (Wang et al., 2018): KCNN is a CNN-based model which concatenates news embedding and knowledge entities embedding to learn the representation of news.

(7) **KLSTM** (Liu et al., 2016;Wang et al., 2018): Similar to KCNN, KLSTM change the CNN module to BiLSTM, which has achieved comptitive results in fake news detection task.

(8) **FB** (Peters et al., 2018): FB denotes the feature-based method of utilizing the pretrain language model for feature extraction. BERT base version is used in the experiments.

(9) **FFT** (Devlin et al., 2018): FFT is the full fine-tune approach based on BERT base.

(10) **KAN** (Dun et al., 2021): KAN is a knowledge-aware attention network which incorporates external knowledge entities through attention mechanisms to predict the veracity of news articles.

(11) **KPL** (Jiang et al., 2022): KPL is the state-of-art model which designs one prompt template and incorporates external knowledge entities into the prompt representation.

### 4.4 Experimental Results

Experiments are conducted in both few-shot and full-scale settings on two datasets. Baseline models are divided into five categories: traditional statistical methods (i.e., DTC, RFC, SVM), neural network methods without external knowledge (i.e., TextCNN, BiLSTM), neural network methods with external knowledge (i.e., KCNN, KLSTM, KAN), pre-trained language model methods without external knowledge (i.e., FB, FFT), and pre-trained language model methods with external knowledge (i.e., KPL).

The experimental results are presented in Table 2. We draw some conclusions in the few-shot setting. First, although the performance is not stable when the number of training data varies, we observe that the increase of training data usually gives rise to the improvement. Second, the deep learning methods without external knowledge are usually worse than the statistical methods. The cause may be that

| Data | Method | Few shot | | | | | Full scale |
|------|--------|------|------|------|------|------|------------|
| | | 2 | 4 | 8 | 16 | 100 | |
| PolitiFact | DTC | 43.16 | 50.89 | 57.31 | 55.42 | 74.58 | 73.44 |
| | RFC | 30.43 | 48.37 | 38.28 | 69.66 | 82.73 | 82.24 |
| | SVM | 24.63 | 47.20 | 51.13 | 60.91 | 84.90 | 86.29 |
| | TextCNN | 20.91 | 19.88 | 31.91 | 51.44 | 75.72 | 84.34 |
| | BiLSTM | 30.36 | 29.78 | 45.75 | 62.11 | 76.11 | 85.96 |
| | KCNN | 20.89 | 16.24 | 31.43 | 55.71 | 76.80 | 85.40 |
| | KLSTM | 29.43 | 20.44 | 34.96 | 65.48 | 76.60 | 86.41 |
| | KAN | 34.05 | 38.05 | 47.94 | 66.27 | 76.91 | 87.28 |
| | FB | 33.26 | 39.65 | 48.34 | 35.96 | 75.93 | 61.02 |
| | FFT | 38.88 | 20.77 | 47.04 | 39.83 | 84.40 | 87.35 |
| | KPL | **61.25** | **63.92** | **68.34** | **74.60** | 83.51 | 89.52 |
| | **KAPALM(Ours)** | 52.78 | 52.66 | 55.49 | 71.74 | **85.77** | **91.34** |
| Gossipcop | DTC | 25.62 | 32.40 | 33.02 | 34.35 | 40.90 | 50.82 |
| | RFC | 33.88 | 30.49 | 34.04 | 37.93 | 49.28 | 43.73 |
| | SVM | 33.69 | 38.34 | 40.20 | 44.66 | 54.36 | 65.03 |
| | TextCNN | 23.68 | 21.22 | 19.69 | 16.84 | 36.26 | 66.82 |
| | BiLSTM | 26.06 | 30.76 | 37.58 | 33.36 | 41.43 | 66.78 |
| | KCNN | 23.31 | 19.75 | 19.12 | 15.84 | 38.26 | 66.51 |
| | KLSTM | 26.54 | 27.76 | 20.45 | 20.84 | 42.25 | 65.94 |
| | KAN | 29.30 | 31.98 | 33.56 | 35.94 | 43.57 | 67.35 |
| | FB | 32.96 | 37.16 | 34.40 | 33.23 | 52.98 | 58.81 |
| | FFT | 34.92 | 40.05 | 39.05 | 30.67 | 58.79 | 70.71 |
| | KPL | 37.80 | 38.78 | 40.20 | 41.63 | 51.72 | 69.20 |
| | **KAPALM(Ours)** | **42.26** | **44.58** | **44.69** | **44.51** | **60.03** | **71.68** |

Table 2: Comparsion with existing models.These scores refer to the F1 scores (%) of fake news.

| Data | Method | Full scale |
|------|--------|------------|
| PolitiFact | **Ours** | **91.34** |
| | w/o GP | 90.05 |
| | w/o IN | 89.72 |
| Gossipcop | **Ours** | **71.68** |
| | w/o GP | 70.99 |
| | w/o IN | 71.02 |

Table 3: Ablation experimental results. "w/o GP" means removing attentive graph pooling layer. "w/o -IN" means that we remove interaction node part.

| Data | Method | Full scale |
|------|--------|------------|
| PolitiFact | FB+Graph | 87.27 |
| | FFT+Graph | 90.05 |
| | Ours(AT+Graph) | **91.34** |
| Gossipcop | FB+Graph | 50.99 |
| | FFT+Graph | 71.53 |
| | Ours(AT+Graph) | **71.68** |

Table 4: Comparison with different fine-tuning methods. "FB+Graph" means that training our model by keeping parameters of pre-train language model frozen and updating other parameters only. "FFT+Graph" means training our model by full fine-tuning all parameters of the model. "AT+Graph" means training our model by adapter tuning just a small amount of additional parameters.

deep learning methods have more parameters than statistical methods and are easy to be over-fitted by the lack of training data. Third, neural network methods with external knowledge are better than those without external knowledge. The results demonstrate that knowledge integration can alleviate the over-fitting problem to some extent. Four, the pre-trained language models can improve the effectiveness of fake news detection generally. Our model is better than KPL in most cases, but wrose than KPL in the few-shot setting on the PolitiFact dataset.

In the full-scale setting, our method outperforms all baselines on two datasets and achieves the highest F1 score. Specifically, KPL is the recent state-of-art model that incorporates large-scale pre-

| Data | Method | Full scale |
|------|--------|-----------|
| PolitiFact | Fine-grained | 90.88 |
| | Coarse- and Fine-grained | **91.34** |
| Gossipcop | Fine-grained | 71.48 |
| | Coarse- and Fine-grained | **71.68** |

Table 5: Ablation experimental results. : "Fine-grained" denotes that the pruned entity graph are fed into both GAT layer and attentive graph pooling layer. "Coarse- and Fine-grained" means that the original entity graph is fed into GAT layer while the pruned entity graph is fed into attentive graph pooling layer.

trained models and external knowledge, and our model is better than KPL with +1.82 and +2.45 improvement on PolitiFact and Gossipcop datasets respectively. Thus, our model can effectively adopt both external knowledge and pre-training language model. In addition, neural network methods with external knowledge are superior to those without external knowledge. It suggest that knowledge can also improve the effectiveness of fake news detection when the training data is sufficient.

### 4.5 Analysis

We conduct a detailed analysis from different perspectives to demonstrate the effectiveness of the modules in our model for fake news detection.

#### 4.5.1 Ablation study

We conduct the ablation study to validate the effectiveness of the coarse- and fine-grained knowledge in our approach. Table 3 presents the experimental results of the models on two datasets. The coarse-grained knowledge is represented by the interaction node in the entity graph while the fine-grained knowledge is provided by the attentive graph pooling layer. If the attentive graph pooling layer is eliminated from our model, the $F1$ drops by 1.34 on the PolitiFact dataset, and $F1$ decreases by 0.69 on the Gossipcop dataset. When the interaction node is removed from our model, the performance of our model declines by 1.6 $F1$ on the PolitiFact dataset and by 0.66 $F1$ on the Gossipcop dataset. Therefore, both the coarse-grained knowledge and fine-grained knowledge are important for our model and able to improve the effectiveness of fake news detection independently.

To validate the contribution of adapting tuning, we conduct an ablation study experiment on two datasets. The results listed in Table 4 demonstrate the superiority of adapter tuning. When the adapter turning method is changed to Full Fine-

Tuning (FFT) or Feature-Based (FB), our model is worse than "FB+Graph" and "FFT+Graph" on two datasets. Our model can benefit from adapter tuning.

We conduct the ablation study to validate the necessity of pruning the entity graph before feeding into the attentive graph pooling layer. We feed the pruned entity graph into both GAT layer and the attentive graph pooling layer. As Table 5 demonstrated, the performance of using both the coarse- and fine-grained knowledge is better than just using fine-grained knowledge on two datasets. Thus, coarse-grained knowledge representations are beneficial to fake news detection.

## 5 Conclusion

In this paper, we propose the Knowledge grAPh enhAnced Language Model (KAPALM) for fake news detection. The proposed model integrates coarse- and fine-grained representations of entity knowledge. Entity graph is used to enrich the knowledge representation of the news article. The entity graph is pruned and fed into the attentive graph pooling layer to represent the fine-grained knowledge. The coarse- and fine-grained knowledge representations extracted from large-scale knowledge graph are combined for improving the fake news detection. Experimental result on two benchmark datasets have shown that the proposed KAPALM outperforms the state-of-the-art baseline models. In addition, KAPALM is able to obtain the competitive performance in the few-shot setting. In future work, we will investigate other effective approaches to mine the accurate knowledge from knowledge graph for fake news detection.

## 6 Limitations

One limitation of our model is that the constructed entity knowledge graph fails to consider the multiple types of relationships or extra attribute information of entities and relationships. If the veracity of a news article is associated with the attributes of the entities or the relationships among entities in this article, it may lead to the poor performance. In addition, although graph neural networks and attentive graph pooling layers can improve the performance in fake news detection, there is still a lack of the Interpretability for our model.

## Acknowledgements

This work was partially supported by the NSFC-General Technology Joint Fund for Basic Research (No. U1936206, U1936105), the National Natural Science Foundation of China (No. 62372252, 62172237, 62077031, 62176028, 62302245), Ministry of Education of the People's Republic of China Humanities and Social Sciences Youth Foundation (No. 63232114). We thank the AC, SPC, PC and reviewers for their insightful comments on this paper.

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
