# OpenReview forum: "KAPALM: Knowledge grAPh enhAnced Language Models for Fake News Detection"
_EMNLP/2023/Conference — EMNLP 2023 Findings_

### Official Review · Reviewer_nxyk · 2023-08-03

**Soundness:** 2

**Excitement:**

1: Poor: I cannot identify the contributions of this paper, or I believe the claims are not sufficiently backed up by evidence. I would fight to have it rejected.

**Paper Topic And Main Contributions:**

The authors propose a knowledge graph enhanced language model (KAPALM) that fuses coarse- and fine-grained representations of entity knowledge from knowledge graphs to conduct fake news detection. They design a method to prune subgraphs, and then use attention mechanism to obtain knowledge representation. By combining it with a textual representation, KAPALM could further improve the detection performance.

**Reasons To Accept:**

•This work copes with the fake news classification task, which is useful in practice.

•The author proposed an effective model for the fake news detection task and verified its effectiveness on the public test set

**Reasons To Reject:**

•The motivation stated in the abstract is not solid enough, and the method proposed in the paper is not novel enough. For the statement " majority of these methods focus on news entity information and ignore the structured knowledge among news entities" in the abstract, some existing methods already do this, like CompareNet[1].

•The comparison of the proposed method with the baseline is not fair enough. The Bert-base-with-adapter used in the article is inconsistent with the Roberta used by KPL, and it is difficult to prove that KAPALM is more effective than KPL. What is the motivation for Bert-base-with-adapter?

•The coarse-grained and fine-grained knowledge mentioned in the article is not clear enough. The knowledge pruning method proposed in the article is intuitive.

[1]. Hu L, Yang T, Zhang L, et al. Compare to the knowledge: Graph neural fake news detection with external knowledge[C], in ACL2021.


**Reproducibility:**

3: Could reproduce the results with some difficulty. The settings of parameters are underspecified or subjectively determined; the training/evaluation data are not widely available.

**Reviewer Confidence:**

4: Quite sure. I tried to check the important points carefully. It's unlikely, though conceivable, that I missed something that should affect my ratings.

---

> ### Author Rebuttal · Authors · 2023-08-29
>
> Reviewer#3: We thank the reviewer for the careful comments. We would like to make some clarifications as follows:
>
> Q1: About reason for using Bert instead of Roberta .
> A1: Roberta is a more finely-tuned version of Bert, so in terms of training, the Roberta model is more powerful than the Bert model. If the performance of our model using Bert exceeds that of using Roberta, it can demonstrate the advantage of our model.Using a Bert model with adapters has two reasons. Firstly, adapters can significantly reduce the time and space costs of training. Secondly, only Bert-base-with-adapter has a readily available library of pre-trained models that can be used. However, we have not found a pre-trained library for Roberta with adapters.

---

### Official Review · Reviewer_oZm4 · 2023-08-04

**Soundness:** 2

**Excitement:**

2: Mediocre: This paper makes marginal contributions (vs non-contemporaneous work), so I would rather not see it in the conference.

**Missing References:**

You should include references that represent the state of the art in respect to whatever dataset(s) you have chosen.

Make sure you include peer-reviewed versions of a paper where this is appropriate (e.g. the BERT paper).

**Paper Topic And Main Contributions:**

Identifying fake news is a highly relevant topic and one that is at the core of NLP given the importance that written text contributes to solving the task. As such it presents a great topical match for EMNLP.

The authors argue that identifying entities in some text and incorporating relationships between entities drawn from some external knowledge graphs might help in this classification task. The architecture that is being proposed is one that generates embeddings from three different angles: the actual text, an entity graph based on entities in the text and related entities in some external graph, a pruned version of this graph. The three embeddings resulting from this process are concatenated before being fed into a binary classifier.

Experiments on the FakeNewsNet dataset are reported.


**Questions For The Authors:**

(1) It is unclear to me that text properties alone (together with what text properties of external sources add to that) would be sufficient to distinguish fabricated news from real news. This may be true for some datasets but I imagine that with more effort put into creating stories (and more sophisticated tools such as those from the GPT family) it would be impossible to distinguish the two classes without incorporating more signals (such as contextual features). What is the underlying intuition in your work? Do you argue that even with the widepread adoption of ChatGPT et al. it will be possible to distinguish fake news from real news based on text (+ entities) alone?

(2) Given you adopt a dataset that includes many more signals I wonder why you do not include all these and THEN try to push forward the state of the art by going beyond what you can achieve. What is the reason to only try to improve on the text classification alone?

(3) Is there a need to include all the baselines? I have the impression that none of 1 - 7 are actually needed as we can assume for BERT to be a stronger baseline (it is also not a surprise that the results demonstrate yet again that traditional methods do not appear to be competitive with neural methods).

(4) Why did you not include a discussion of ethical issues? Given the topic this would have been a helpful addition.

**Reasons To Accept:**

* This is a highly relevant research area, and any work that pushes our understanding as to how to identify misinformation of any form is a step forward.

* The work is conducted on a well-known benchmark dataset and is therefore easily contextualisable with other work in the field.

* Conducting an ablation study is a strong aspect of the experimental work as it offers more nuanced insights into what components of the architecture contribute how much  to the overall performance.

**Reasons To Reject:**

There are a number of weaknesses in this work. Among the biggest issues I consider the following:

* Weak baselines: Given the chosen dataset contains much more information than the textual content I would want to see the results of this work compared to the state of the art reported in the literature that looks at the same dataset. To pick just one recent example [1], the results reported (such as F1) appear to be way higher than anything in Table 2 (and that is also true for the baselines reported in [1]). What is needed is a comparison against what the current state of the art is as reported in the literature (ideally reproduced to conduct significance tests where appropriate).

* Only a single dataset is used to explore the problem (well, it is two different parts but in the end it is one fairly specific dataset). There are many more benchmark datasets for text classification (including fake news detection) that could be included to provide more confidence in the findings.

* There are many missing details (and no supplementary material such as code) making it impossible to replicate the work. For example, unless I have overlooked it I cannot see what knowledge graphs are actually being used.

* There are no statistical significance tests (and terms such as "outperform" should therefore not be used)




[1] Donabauer "Exploring Fake News Detection with Heterogeneous Social Media Context Graphs". ECIR 2023.

**Reproducibility:**

2: Would be hard pressed to reproduce the results. The contribution depends on data that are simply not available outside the author's institution or consortium; not enough details are provided.

**Reviewer Confidence:**

4: Quite sure. I tried to check the important points carefully. It's unlikely, though conceivable, that I missed something that should affect my ratings.

**Typos Grammar Style And Presentation Improvements:**

The paper needs to be properly proofread. It has many language problems throughout.

---

> ### Author Rebuttal · Authors · 2023-08-29
>
> Reviewer#2:We thank the reviewer for the deeper thought about the property of the union graph and appreciate your recognition about the contribution of the property.
> Q1：About weak baseline.
> A1: We did not use contextual information. The article you mentioned used contextual information, which is why its results are higher than ours. Our comparison with similar methods (using only news context and external knowledge) can demonstrate the superiority of our model.
>
> Q2: About dataset.
> A2: In fact, Politicact and Gossipcop are two completely different datasets rather than one dataset.
>
> Q3: About missing details.
> A3: I will upload the code for constructing the knowledge graph and all related codes to Github for public disclosure so that people can reproduce my work. The process of constructing a knowledge graph is detailed in sections 3.3, 3.4, and 3.5 of the paper.
>
> Q4: About lacking statistical significance tests.
> A4: We will supplement with statistical significance tests in subsequent work.
>
> Q5: About the necessary to include all baseline.
> A5: Our method is mainly used to compare with the KPL model (a strong baseline in this field). As the latter conducted 1-7 experiments, in order to maintain consistency with his experiments, I set 1-7 as the baseline model.
>
> Q6: About underlying intuition and ChatGPT challenge.
> A6: Our underlying intuition is that news content is highly condensed and comprised of a large number of entity which are useful to detect fake news. About ChatGPT challenge, my answer is yes. My model can detect fake news by using text and external knowledge.
>
> Q7: About the reason why we do not include all signals.
> A7: The dataset does not contain contextual information, so we cannot obtain other relevant signals.
>
> Q8: About ethical issues.
> A8:We will supplement with ethical issues in subsequent work.

---

### Official Review · Reviewer_4XMg · 2023-08-11

**Soundness:** 3

**Excitement:**

3: Ambivalent: It has merits (e.g., it reports state-of-the-art results, the idea is nice), but there are key weaknesses (e.g., it describes incremental work), and it can significantly benefit from another round of revision. However, I won't object to accepting it if my co-reviewers champion it.

**Paper Topic And Main Contributions:**

The paper investigate to integrate structured knowledge among entities in the news article to enhance the fake news detection. The proposed method outperforms previous KG related approaches that do not utilise structured information. It also shows promising results in the few-shot setting.

**Questions For The Authors:**

Did you conduct error analysis to explore the reason why your proposed method is worse than KPL in the few shot setting on Politifact, but better than KPL on Gossipcop?

**Reasons To Accept:**

The approach is novel and achieves SOTA on two datasets with full scale training, and show promising performance in the few-shot setting.

**Reasons To Reject:**

Lack statistical significance tests.

**Reproducibility:**

4: Could mostly reproduce the results, but there may be some variation because of sample variance or minor variations in their interpretation of the protocol or method.

**Reviewer Confidence:**

3: Pretty sure, but there's a chance I missed something. Although I have a good feel for this area in general, I did not carefully check the paper's details, e.g., the math, experimental design, or novelty.

---

> ### Author Rebuttal · Authors · 2023-08-29
>
> Reviewer#1: Thanks for your valuable and careful comments. We will explain you concerns point by point.
> Q1: About lacking statistical significance tests.
> A1: We will supplement with statistical significance tests in subsequent work.
>
> Q2: About lacking error analysis.
> A2: Because of the difference between the two datasets. The average number of entities per news in the Politifact is 55, while the average number of entities per news in the Gossipcop is 29. In few shot setting, due to the lack of samples, the noise generated by entities in the Politifact have greater impact on the prediction results, resulting in my proposed method being worse than the KPL model in a few shot setting on the Politifact, but better than the KPL model on the Gossipcop.

---

### Meta-Review · Area_Chair_NDoN · 2023-09-20

**Recommendation:** 2

**Metareview:**

The paper presents a method to combine entity knowledge from KGs with language models for imporving the performance on the task of fake news detection, as measured on two benchmark data sets. Entity mentions are extracted from text and linked to a knowledge graph, then a subgraph is extracted and used in a graph-neural network, and its output combined with a representation of text obtained using a Transformer-based model.

All reviewers appreciated the results and the introduction of a new method that is effective on standard benchmark data sets. One reviewer also appreciated the ablation experiments, which provided additional insights in the results.

Two reviewers mentioned needing to add statistical significance results on the experiments and also having more baseline methods that use information from Knowledge graphs and a more detailed discussion on part work that did this. Doing so would have likely increased the soundness scores.

All reviewers acknowledged to reading the author response, but did not change their scores as a result of it.

---

### Decision · Program_Chairs · 2023-10-07

**Decision:**

Accept-Findings

**Comment:**

The paper presents a method to combine entity knowledge from KGs with language models for imporving the performance on the task of fake news detection, as measured on two benchmark data sets. Entity mentions are extracted from text and linked to a knowledge graph, then a subgraph is extracted and used in a graph-neural network, and its output combined with a representation of text obtained using a Transformer-based model.

All reviewers appreciated the results and the introduction of a new method that is effective on standard benchmark data sets. One reviewer also appreciated the ablation experiments, which provided additional insights in the results.

Two reviewers mentioned needing to add statistical significance results on the experiments and also having more baseline methods that use information from Knowledge graphs and a more detailed discussion on part work that did this. Doing so would have likely increased the soundness scores.

All reviewers acknowledged to reading the author response, but did not change their scores as a result of it.